# Identification of a Novel Renal Metastasis Associated CpG-Based DNA Methylation Signature (RMAMS)

**DOI:** 10.3390/ijms231911190

**Published:** 2022-09-23

**Authors:** Jürgen Serth, Inga Peters, Olga Katzendorn, Tu N. Dang, Joana Moog, Zarife Balli, Christel Reese, Jörg Hennenlotter, Alexander Grote, Marcel Lafos, Hossein Tezval, Markus A. Kuczyk

**Affiliations:** 1Department of Urology and Urologic Oncology, Hannover Medical School, 30625 Hannover, Germany; 2Department of Urology, University of Tübingen, 72076 Tübingen, Germany; 3Department of Neurosurgery, University Hospital Gießen und Marburg, 35043 Marburg, Germany; 4Department of Pathology, Hannover Medical School, 30625 Hannover, Germany

**Keywords:** renal cell carcinoma, metastasis, NKX6-2, NHLH2, INA, THBS4, hypermethylation, signature, prognosis, CpG-methylation, DNA-methylation

## Abstract

Approximately 21% of patients with renal cell cancer (RCC) present with synchronous metastatic disease at the time of diagnosis, and metachronous metastatic disease occurs in 20–50% of cases within 5 years. Recent advances in adjuvant treatment of aggressive RCC following surgery suggest that biomarker-based prediction of risk for distant metastasis could improve patient selection. Biometrical analysis of TCGA-KIRC data identified candidate loci in the NK6 homeobox 2 gene (*NKX6-2*) that are hypermethylated in primary metastatic RCC. Analyses of *NKX6-2* DNA methylation in three gene regions including a total of 16 CpG sites in 154 tumor-adjacent normal tissue, 189 RCC, and 194 metastatic tissue samples from 95 metastasized RCC patients revealed highly significant tumor-specific, primary metastatic-specific, and metastatic tissue-specific hypermethylation of *NKX6-2*. Combined CpG site methylation data for *NKX6-2* and metastasis-associated genes (*INA*, *NHLH2*, and *THBS4*) demonstrated similarity between metastatic tissues and metastatic primary RCC tissues. The random forest method and evaluation of an unknown test cohort of tissues using receiver operator characteristic curve analysis revealed that metastatic tissues can be differentiated by a median area under the curve of 0.86 (*p* = 1.7 × 10^−8^–7.5 × 10^−3^) in 1000 random runs. Analysis of variable importance demonstrated an above median contribution for decision-making of at least one CpG site in each of the genes, suggesting superior informativity for sites annotated to *NHLH2* and *NKX6*-2. Thus, DNA methylation of *NKX6-2* is associated with the metastatic state of RCC tissues and contributes to a four-gene-based statistical predictor of tumoral and metastatic renal tissues.

## 1. Introduction

Renal cell carcinoma (RCC) occurs as the sixth most frequent cancer in men and tenth most frequent cancer in women, with increasing incidence [1]. Approximately 20% of patients present with metastatic disease at the time of diagnosis, whereas disease recurrence and metachronous metastasis following surgery is observed in 20–50% of patients within 5 years [2,3]. Although new therapies have improved the prognosis of metastatic disease, the 5-year survival of affected patients is still limited [4]. Therefore, reducing the risk of developing metastases is of primary interest, and the recent approval of an adjuvant treatment regimen using the PD-1 antibody pembrolizumab in high-risk RCC represents the first therapeutic answer to this treatment challenge [5]. Inclusion criteria for patients currently rely on clinical and histological parameters or scoring systems, such as the University of California Integrated Staging System (UISS), but limitations in the prognostic accuracy of these models underline the need for additional prognostic biomarkers to improve individual clinical predictions and permit personalized therapeutic strategies and selection for adjuvant therapies [6,7,8].

The understanding of RCC has evolved substantially due to the comprehensive molecular characterization of RCC provided by The Cancer Genome Atlas (TCGA) project, including detailed information about genetic and epigenetic alterations in the most common histological subtype, clear cell renal carcinoma ccRCC (KIRC database) [9]. Interestingly, data show that somatic mutations other than VHL alterations are relatively rare in sporadic RCC. Moreover, a limited association or low informativity of genetic alterations with clinicopathological parameters or RCC prognosis has been observed; thus, the translational clinical use of genetic mutation profiles seems to be questionable [8,9,10]. In contrast, epigenetic alterations, such as DNA hypermethylation, frequently followed by subsequent epigenetic silencing of tumor suppressor genes, has been demonstrated to occur in RCC and to be associated with clinical or histopathological parameters and the clinical course of RCC, suggesting epigenetic alterations as promising biomarker candidates in RCC [10,11,12,13,14]. Numerous investigations, including our own work, have revealed associations of hypermethylation and adverse histopathological characteristics [15,16,17,18,19,20] with metastatic disease [15,16,17,18,20,21], as well as associations with survival [15,16,18,20,22,23,24,25,26] or the therapeutic response [27,28].

Although alterations in DNA methylation usually affect a comparatively large proportion of respective tumors, clinical translation of potential markers is still hampered by non-informative cases. Our recently published association of the DNA methylation of CpG loci of *INA*, *NHLH2*, and *THBS4* with metastasis revealed a subset of tissues showing no alterations in the analyzed loci. Therefore, it seems rational that the translational use of DNA methylation for the prediction of tumor behavior would be carried out using marker panels [29]. A five CpG-methylation-based signature and a four-gene-based signature have been suggested to predict overall survival independent of clinical parameters or cancer-specific survival, respectively [26,30]

Here, we investigated whether detection of metastatic renal tissues using CpG-methylation of *INA*, *NHLH2*, and *THBS4* can be improved by including the methylation information of biometrically-identified candidate loci in *NKX6-2*. NKX6-2 is a transcription factor with a highly conserved DNA-binding domain, the homeobox, and is involved in neuronal and pancreatic endocrine development [31,32]. DNA hypermethylation of *NKX6-2* has been detected in lung adenocarcinoma, colorectal carcinoma, and bladder carcinoma [33,34,35]. Notably, it has been described as part of a biomarker panel for early bladder cancer detection in urine [33], and methylome analysis in RCC has suggested that hypermethylation of *NKX6-2* is associated with more aggressive RCC, demonstrating shortened overall survival [36].

Using an extended investigation approach (Figure 1) we show that CpG sites in different regions of *NKX6-2* present hypermethylation, in both metastasized primary RCC tissue and RCC-derived distant metastatic tissues. Moreover, specific CpG sites in *NKX6-2* provide an above average contribution to the diagnostic signature for efficient and robust detection of metastatic risk by combined use of *INA*, *NHLH2*, *NKX6-2*, and *THBS4* CpG-site methylation.

## 2. Results

### 2.1. In Silico Identification of the Association of NKX6-2 Loci Methylation and State of Distant Metastasis

Univariate logistic regression of TCGA KIRC methylation data from 282 tumor tissues, including 232 M0 and 52 M+ tumors, identified three loci annotated for *NKX6-2* (cg06082548, cg01384488, and cg19701540) that were associated with the state of distant metastasis in patients. Ranking among the top 30 candidate loci, the observed mean fold changes in methylation for M0 vs. M+ primary tumor comparisons were 1.56, 1.84, and 1.94, with corresponding *p*-values of 4.36 × 10^−19^, 3.46 × 10^−23^, and 1.2 × 10^−21^ (Bonferroni-Hochberg corrected).

### 2.2. Evaluation of NKX6-2 Candidate Loci in Primary RCC, RCC-Associated Metastatic Tissues, and Cell Models

Following the set-up of three pyrosequencing assays covering or positioned as close as possible to the candidate loci (Figure 2, Table 1), we observed high relative methylation in a large part of the cell models, including representatives of renal, urothelial, prostatic, and mammary cancers, for all of three *NKX6-2* regions (R3-R1) analyzed, whereas the relative methylation in normal primary cells was low (Figure 3).

Comparative analysis of primary TU and paired adN samples demonstrated significant tumor-specific hypermethylation (all *p* < 1.25 × 10^−14^) for all three regions (Figure 4a, Table 2). M+ tumors also demonstrated hypermethylation (all *p* ≤ 0.008) compared with the M0 tumors (Figure 4b, Table 2). Eventually, the comparison of M0 and Mtx tissues revealed significant hypermethylation (all *p* ≤ 1.04 × 10^−5^) for the metastases in each *NKX6-2* region R3-R1 (Figure 4c, Table 2).

### 2.3. Similar CpG-Specific Methylation in Metastatic Primary Tumor Tissues and Metastatic Tissues

Whether and to what extent DNA-methylation is similar in metastatic primary cancers and distant metastatic tissue samples is an important question for both the basic understanding of changes in cells escaping the primary cancer and translational diagnostic approaches aimed at predicting or detecting metastatic alterations in RCC. In addition to the *NKX6-2* findings above and recently published analogous results identifying gene-wise averaged methylation in *INA*, *NHLH2*, and *THBS4* that are associated with the metastatic potential of tissues, we analyzed whether a similarity in the methylation in these four genes is observed when evaluating CpG-specific alterations [20]. First, CpG-specific methylation was analyzed in bivariate logistic regression comparing M0 and Mtx tissues. The forest plot of odds ratios (ORs) showed that all loci of the four candidate genes appeared to be associated with increased methylation and increased risk of metastatic tissue classification, providing potential information for molecular detection of the metastatic state (Appendix A). Moreover, CpG-site specific comparisons of methylation in M0 and M+ tissues revealed significant differences in 9 of 16 CpG sites (56%), but no effect of the covariate age (Appendix A).

To investigate whether the combined methylation information from all measured loci allows discrimination of tissue samples with different metastatic potential, we performed unsupervised clustering analysis of all primary tumors and metastatic tissue samples, revealing two stable clusters. Cluster 2 included samples of low to medium average methylation over all CpG sites (Figure 5a), whereas cluster 1 exhibited medium to high average methylation, and a large part comprised metastases (Figure 5a, orange boxes) or metastasized primary tumor tissues (Figure 5a, red boxes). In contrast, cluster 2 presented a mixture of localized primary tumors (Figure 5a, blue boxes) and M+ or Mtx tissues (Figure 5a, orange and red boxes). Moreover, it seems obvious that some cases had few differences in individual CpG methylation of M+ and Mtx tissue samples. This observation is supported by statistical multidimensional analysis (MDA, Figure 5b) and PCA (Figure 5c). Both methods indicate that a large portion of M+ and Mtx samples fell into widely overlapping areas, with the exception of two outlier Mtx samples.

### 2.4. Development and Evaluation of a RMAMS

Considering that statistical analysis of the similarities in M+ and Mtx tissues demonstrated considerable congruence in the methylation of the *INA*, *NKX6-2*, *NHLH2*, and *THBS4* loci, we asked whether this could be exploited for molecular identification of the metastatic state of tissue samples. Statistical learning using a random forest classification model demonstrated that, following training of the model and test cohort, a statistically untrained independent subset of M0 and Mtx tissues could be diagnosed with high average diagnostic efficiency, as indicated by ROC-AUC analysis.

A 1000-fold repetition using random subsets for training and test cohorts demonstrated a distribution of the resulting ROC-AUC values as shown in the box plot in Figure 6a. The median AUC was 0.86 for the comparison of M0 vs. Mtx. Notably, both the distribution and median AUC showed no significant changes when increasing the number of repetitions to a maximum value of 5000 (data not shown). An exemplary ROC curve analysis showing the results of a single random data split matching the median AUC value of 0.86 for detection of Mtx tissues in all random runs for an unknown test cohort is presented in Figure 6b. In this particular permutation, 46 cases were classified, showing 20 true positives, 27 true negatives, 5 false positives, and 4 false negatives (sensitivity = 0.83, specificity = 0.77). These values corresponded to a positive likelihood ratio (PLR) of 3.67, a negative likelihood ratio (NLR) of 0.22, and a diagnostic odds ratio of 17, indicating good diagnostic efficiency of the random forest classifier, overall. Evaluation of the complete random data set for a surrogate confidence interval demonstrated that 95% of ROC-AUC values fell within the interval of 0.73–0.94, corresponding to *p*-values of 1.7 × 10^−8^–7.5 × 10^−3^.

The analysis of variable importance was also carried out for the complete random data set of 1000 permutations and showed that the methylation information for at least one of the CpG sites of each analyzed gene contributed above median importance to the random forest classification model, also pointing to a possible prominent role of *NKX6-2* R2 CpG sites in classification (Figure 6c). Interestingly, additional evaluation of the random forest classifier by predicting tissue classes in another independent test cohort consisting of M0 and M+ tissues also demonstrated diagnostic informativity in the ROC-AUC analysis, with a median AUC of 0.71 (95% of ROC-AUC = 0.69–0.73, *p* = 3.410^−4^–2.7 × 10^−3^). Notably, application of the classifier for the discrimination of M0 and M+ samples (i.e., a classification task the classifier was not trained for) revealed a stable and narrow distribution of AUC values (Figure 6a, M0/M+).

## 3. Discussion

The understanding and prediction of metastasis of primary cancers, such as RCC, is of great interest considering the potential implications for the development of molecular therapeutic options, molecular diagnostics and prognostics, as well as the stratification of patients for individualized and/or adjuvant treatments. Therefore, molecular-based improvement of classical clinicopathological selection of patients for modern therapies could both increase the proportion of those who would likely benefit from neoadjuvant or adjuvant strategies, and reduce therapy-associated toxicity by avoiding medication in unselected patients.

To identify candidate loci for metastasis-specific DNA hypermethylation, we used in silico analysis of the KIRC data published by the TCGA consortium and subsequent validation using primary RCC and RCC metastases tissue cohorts. Analogous to previous results revealing an association of methylation in *INA*, *NHLH2*, and *THBS4* with the metastatic state of tissues [20], we analyzed the relevance of methylation in *NKX6-2* to RCC metastasis and whether combined application of CpG-based methylation information for the four genes provides information that improves the detection of metastatic tissues or metastasized primary cancers.

DNA methylation of *NKX6-2* has been reported to be associated with aggressive RCC as part of a methylation signature for the detection of bladder cancer and to be a target of hypermethylation in other tumor entities [33,34,35,36].

Thus, both our biometrical analysis identifying candidate CpG loci in three regions of *NKX6-2* as being among the top 30 candidates for association with RCC metastasis, and the detection of frequent hypermethylation in tumor cell models representing frequent human tumor entities, are in line with previously published data and point to a broader relevance of alterations in *NKX6-2* methylation.

Analyses of primary RCC not only confirmed tumor-specific hypermethylation, but demonstrated hypermethylation in a comparison of localized and metastasized primary RCC, confirming the results of the in silico analysis of the KIRC data and providing an explanation for previous findings of an association with aggressive cancers. These results were obtained uniformly for each of the three investigated gene regions spanning a genomic region including putative regulation of gene transcription and gene body sequences.

Our comparison of metastatic and localized primary RCC tissues revealed significant hypermethylation of all three *NKX6-2* regions, independently supporting the relevance of *NKX6-2* methylation in RCC metastasis. Therefore, the association of methylation and metastatic disease progression suggests *NKX6-2* as a promising candidate for subsequent targeted functional analysis of RCC metastases.

Considering that the distributions of methylation values obtained for metastasized primary tissues (M+) and metastatic tissue (Mtx) samples appeared to be similar, the question was raised as to whether this characteristic would also be found when CpG-site-specific methylation data are evaluated, and corresponding data for *INA*, *NHLH2*, and *THBS4*, which were previously identified to be statistically associated with RCC metastasis, are included.

Firstly, unsupervised clustering of CpG-specific methylation data for *INA*, ***N****HLH2*, ***N****KX6-2*, and ***T****HBS4* (INNT) revealed the most stable outcome for k-means clustering using two centroids. Visual inspection of clustering on the heatmap presentation of all tissue samples revealed, on average, a highly methylated cluster with significant enrichment of M+ and Mtx tissues. In contrast, the other, low-methylation cluster was made up of a mixture of M0, M+, and Mtx tissues. However, roughly half of the analyzed tissues of this cluster exhibited very low or missing signal for most of the measured CpG sites, indicating that CpG-site methylation in INNT was not informative for approximately 20% of cases. Moreover, in both clusters, M+ and Mtx tissues were observed in direct neighborhood, indicating a similarity in the methylation profiles of both types of tissue. This hypothesis was supported by statistical MDS and PCA equally demonstrating a broad overlap of both tissue types and underlining the similarity of M+ and Mtx samples with respect to the CpG methylation profiles in INNT.

Therefore, we questioned whether the similarity in INNT methylation could be exploited for the definition of an RMAMS permitting detection of tissues with metastatic behavior. Interestingly, statistical learning using the random forest algorithm of a random training cohort of M0 and Mtx tissues allowed classification of the untrained test cohort with a high and robust median diagnostic ROC-AUC value of 0.86 in 1000 random splits of training and test data. Thus, information on the methylation of INNT can be statistically learned and is sufficient for successful classification of a substantial portion of unknown samples. Though random forest classification, as in many other modern statistical learning methods, does not provide exact insight into the final model used for diagnostic decisions, the minimum importance analyses of variables provided information about the most relevant input variables for classification. Using the averaged importance data from all random runs, it appeared that at least one CpG site in each of the analyzed genes belonged to the “high informative” group of predictors, whereas superior informativity was likely obtained from CpG sites in region 2 of *NKX6-2* and all of the *NHLH2* candidate sites in the statistical decision model.

The second, independent evaluation of the classification model based on the prediction of primary tissues for the state of distant metastasis assessed whether molecular similarity between metastatic tissues and metastasized primary tumors allows differentiation in a diagnostic set-up for which the statistical learner was not trained. This evaluation revealed a significant discrimination between tissues with different metastatic potential, showing a low variance of ROC-AUC values in the random runs. Thus, all of our unsupervised and supervised statistical analyses provided strong evidence that M+ and Mtx tissues have a similar pattern of methylation of CpG sites in INNT, allowing RMAMS-based discrimination of samples with different metastatic potential.

Moreover, inspection of diagnostic parameters of RMAMS indicated that inclusion of information on *NKX6-2* CpG-based methylation improved the PLR and NLR achieved by the *INA*, *NHLH2*, and *THBS4* annotated CpG sites, further shifting diagnostic parameters towards values that are characteristic of medical applications [37]. Although significant discrimination of tissues with metastatic characteristics was achieved in both evaluation experiments, ROC-AUC values showed that false positive or negative classifications occur. On the one hand, false negative decisions have to be expected when inspecting the results of an unsupervised cluster analysis showing that a subset of Mtx and M+ samples exhibited very low or absent methylation for INNT CpG sites. However, under the assumption that inclusion of further metastasis-associated candidate CpG sites should allow further minimization of the proportion of yet undetectable (i.e., false negative) tissues, improved diagnostic efficiency, such as observed after combining *NKX6-2* data with our initial methylation signature, should be achievable. On the other hand, assessment of the relevance of false positive classifications revealed a current limitation of our study design.

As no clinical information about late onset of metastasis (metachronous metastasis) in patients was available for the cohort under analysis, whether the false positive tissues were candidates for an undetected or metachronous metastasis or represent a diagnostic flaw, remains unclear. The first aspect would be of substantial clinical interest considering that stratified patients could be subjected to (neo)adjuvant therapy using recent immune checkpoint treatments with manageable side effects. Therefore, although our results likely provide a basis for such molecular stratification, it is evident that appropriate study cohorts are required to experimentally answer this unmet clinical need.

Methylation signatures for the detection of aggressive or high-risk RCC have been suggested in pursuing different clinical endpoints. Wei et al. identified a five-CpG-site methylation-based signature showing an association with overall patient survival independent of clinical parameters, whereas Joosten et al. concluded in her meta-analysis that methylation in five genes can be combined with a clinical scoring system to predict cancer-specific survival of patients [30,38]. Both studies concluded that methylation markers can add prognostic information to routine clinicopathological parameters. Interestingly, three genes of the five-gene-methylation signature, *GATA5*, *LAD1*, and *NEFH*, have been mutually confirmed despite variation in clinical endpoints, measurement of independent cohorts, and application of different methods for the detection of methylation by different research groups [26,27,28,38,39]. Thus, we assume that current development of methylation signatures as a whole is still in the discovery phase as a result of more or less pronounced limitations in discovery and evaluation cohort size and available clinical information. Nonetheless, all of the signature data consistently indicate that the objective of methylation-based risk assessment in RCC patients for personalized treatment can be achieved in principle, even if the way there involves considerable effort to evaluate known and new information in a uniform test cohort.

In conclusion, we demonstrated that CpG-based methylation information from *INA*, *NHLH2*, *NKX6-2*, and *THBS4* has similarities in renal metastatic and primary tissues based on the clinical state of distant metastasis and can be used for statistical prediction of the metastatic potential of renal tissues. Therefore, CpG-based methylation is a candidate for molecular stratification of high-risk patients who will likely benefit from (neo)adjuvant therapy options.

## 4. Material and Methods

### 4.1. Study Design

To identify metastasis-associated candidate loci, we used an in silico analysis of level 3 data from the TCGA KIRC HM450k methylation dataset using statistical software R version 3.6.1 as described previously [9,20,40] (Figure 1A). First, gene-wise averaged CpG site-relative methylation values were used to analyze *NKX6-2* methylation in various human cell line models and tumor-specific hypermethylation, metastasis-associated hypermethylation in primary cancers, and metastatic tissue-specific hypermethylation (Figure 1B). To investigate the similarities in DNA methylation in metastasized primary tumor (M+) and metastatic tissue (Mtx) samples, we compared CpG-specific methylation of *NKX6-2* and previously published candidate loci of *INA*, *NHLH2*, and *THBS4* exhibiting gene-wise averaged methylation values associated with metastatic potential (Figure 1C) [20]. Randomly selected subsets of primary tumor tissues without distant metastasis (M0) and M+, as well as Mtx, samples were used for statistical learning of a renal metastasis associated methylation signature (RMAMS) and evaluated in two independent test cohorts (Figure 1D).

### 4.2. Study Cohort

Characteristics of the patients subjected to methylation analysis of *NKX6-2* are given in Appendix A. Methylation analyses for all CpG sites in the four candidate genes were carried out in a maximum of 189 RCC tumor tissues, 154 paired tumor adjacent normal tissues (adNs), and 194 metastases from 95 patients with metastatic RCC disease. Patient characteristics in regard to the *INA*, *NHLH2*, and *THBS4* genes, description of the common metastatic tissue cohort and tissue sampling, TNM classification, grading, and tissue treatment were described previously [20]. Ethical approval was obtained from the ethical boards of Eberhard Karls University Tübingen and Hanover Medical School (no. 128/2003V and 1213-2011; approved 14 October 2011). Written informed consent was obtained from all patients. The study was performed in accordance with the Helsinki Declaration.

### 4.3. Nucleic Acid Extraction, DNA Bisulfite Conversion, and DNA Methylation Analysis

Histological estimation of tumor cell content, DNA isolation from frozen sections and formalin-fixed paraffin-embedded tissue sample punches, and bisulfite conversion of DNA were achieved as reported elsewhere [15,24]. DNA methylation analyses were carried out by pyrosequencing. PCR reactions and pyrosequencing template preparation were performed as described previously [15,23]. *NKX6-2* pyrosequencing assays were designed by PyroMark Assay Design 2.0 software (Qiagen, Hilden, Germany) and the hg19 genome assembly as provided by the UCSC table browser. Primer sequences, annealing temperatures, and genomic regions are presented in Appendix A. Detailed CpG-site information for biometrical candidate sites and loci covered by pyrosequencing analysis and used for subsequent statistical evaluation are summarized in Table 1. The genomic context of *NKX6-2*, annotated HM450K CpG sites, candidate loci, and sites covered by the pyrosequencing assay are presented in Figure 2.

### 4.4. Statistical Analysis

All statistical analyses were performed in R version 3.6.1 software, R-Studio^®^, and program libraries as specified below [40,41]. Statistical tissue group comparisons were carried out using either gene-wise aggregated methylation values obtained by calculating the corresponding means of CpG-site-specific methylation values or loci-specific methylation values, as summarized in Figure 1. Tumor-specific hypermethylation of paired samples was evaluated by the two-sided paired *t*-test, whereas independent group comparisons to demonstrate the association of methylation and metastatic state were performed by bivariate logistic regression models with age as the covariate. Multiple metastatic tissues were evaluated following patient-wise aggregation and the calculation of mean methylation values.

Unsupervised statistical classification analyses of primary tumor and metastatic tissue samples was carried out by use of loci-specific methylation, including 34 CpG sites annotated to the six candidate regions of *INA* (7 sites), *NHLH2* (4 sites), *NKX6-2* (16 sites), and *THBS4* (7 sites). Missing data for unsupervised clustering analysis were imputed using the mice package for R [42]. K-means clustering with two clusters was identified as the most stable clustering method using Jaccard indices [43]. Heatmap analyses showed consensus clusters after 100 runs using bootstrapped resampling [44]. Similarities in the DNA methylation observed in primary tumors showing distant metastasis and independent metastatic tissue samples were analyzed by multi-dimensional scaling (MDS) and principal component analysis (PCA).

Supervised classification analysis was performed by applying the tidymodels framework for random forest classification without optimizing model parameters and applying analysis of variable importance [45,46]. A total of 1000 runs for random splits of training and test cohorts were carried out. To assess the diagnostic efficiency of the random forest model, receiver operator characteristic (ROC) curve analysis of the two independent test cohort classifications was performed for each random run. The distribution of the results of the receiver operator characteristics and area under curve (ROC-AUC) analyses was summarized as a box plot, and one representative ROC-AUC plot is presented. Mann–Whitney U statistics were used to estimate the significance of the area under the curve being different from the null hypothesis.

## Figures and Tables

**Figure 1 ijms-23-11190-f001:**
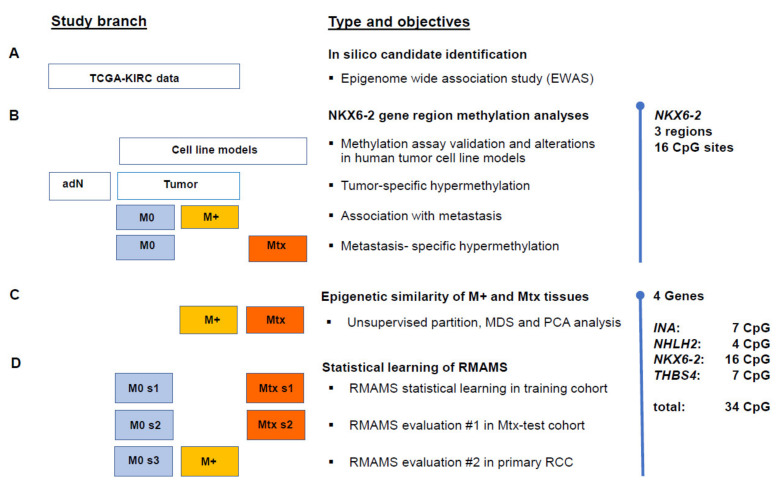
Study design. (**A**) Boxes illustrate data of the TCGA KIRC, cell models (**B**) or renal and metastatic tissues (**C**,**D**). The columns of the boxes represent tumor-adjacent normal renal tissues (adN), primary renal tumor tissues without (M0, blue shading) and with distant metastasis (M+, yellow shading), and renal metastatic tissues (Mtx, orange shading). (**B**) Relevance of *NKX6-2* methylation was investigated using the average of methylation of 16 CpG sites in three gene regions. (**C**,**D**) The epigenetic signatures of primary RCC with distant metastasis were evaluated by combined CpG site-specific methylation of *NKX6-2* and metastasis-associated methylation of *INA*, *NHLH2*, and *THBS4*. Abbreviations used: MDS, multidimensional scaling; PCA, principal component analysis; RMAMS, renal metastasis associated methylation signature; s1–s3, random subsets of tissue groups.

**Figure 2 ijms-23-11190-f002:**
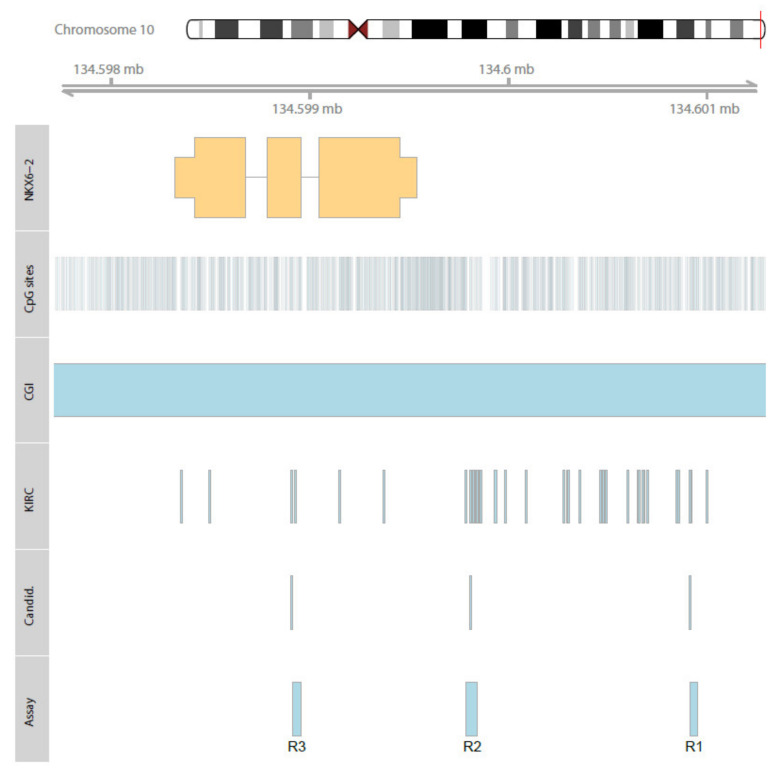
Genomic organization of *NKX6-2*. Localization of exons (higher part of orange rectangles) and the 5′UTR and 3′UTR relative to genomic regions (lower part of orange rectangles), CpG sites annotated for the region (CpG sites), localization of the CpG island (CGI), positions of CpG sites considered in the TCGA KIRC study (KIRC), candidate CpG sites showing an association with the state of distant metastasis.

**Figure 3 ijms-23-11190-f003:**
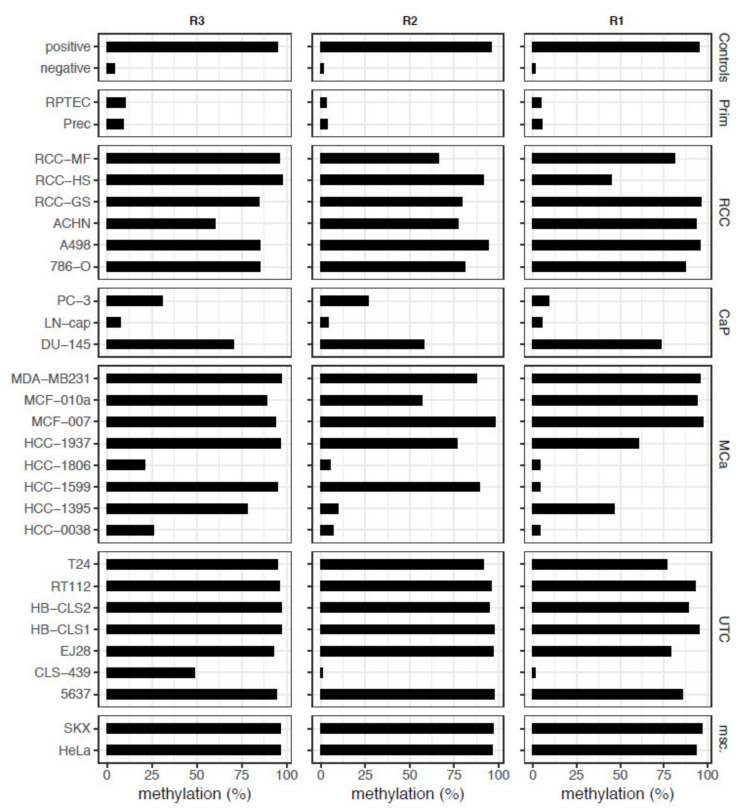
*NKX6-2* methylation in regions R3-R1 in cell models. Pyrosequencing analysis provided relative methylation in the *NKX6-2* R3-R1 regions in % of control DNA (Controls), normal primary cells (Prim), and tumor cells representing renal cell cancer (RCC), prostate cancer (CaP), mammary cancer (MCa), urothelial cancer (UTC), and miscellaneous (msc.) models.

**Figure 4 ijms-23-11190-f004:**
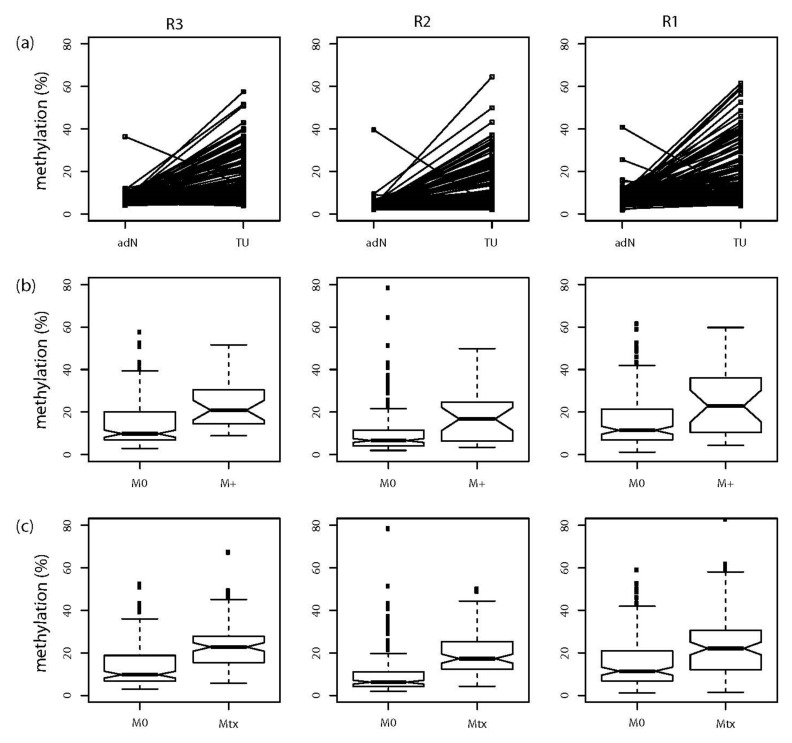
Tumor-specific, state of distant metastasis-specific, and metastatic tissue-specific hypermethylation in NKX6-2 regions R3-R1. (**a**) Strip chart presentation of hypermethylation analysis of regional methylation in NKX6-2 R1-R3 in paired adjacent normal tissue (adN) and renal tumor (TU) tissues. Corresponding statistical analysis showed significant tumor-specific hypermethylation for the R1, R2, and R3 regions (all *p* < 1.25 × 10^−14^, for statistical results see Table 2). (**b**) Box plot of average methylation in regions R3-R1 in NKX6-2 in tumors without (M0) and with metastatic disease (M+). Medians, notches showing the estimated confidence interval, 25% and 75% quartiles, whiskers indicating the 99.3% interval (two-sided 1.5-fold of interquartile range), and outliers (black squares) of the relative methylation distributions are shown. All *p* ≤ 0.008 (for statistical results see Table 2). (**c**) Box plot of methylation in NKX6-2 regions R1–R3 comparing localized primary tumor tissue (M0) and metastatic tissue samples (Mtx). All regions demonstrate metastatic tissue-specific hypermethylation (all *p* ≤ 1.04 × 10^−5^, for statistical results see Table 2). Box plot presentation as described in (**b**).

**Figure 5 ijms-23-11190-f005:**
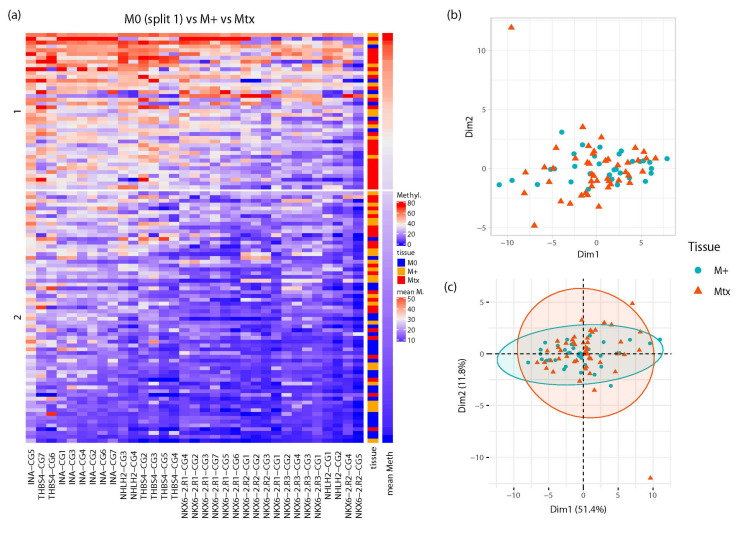
Similarity of *INA*, *NHLH2*, *NKX6-2*, and *THBS4* CpG methylation patterns in metastasized primary tumors and metastatic renal tissues. (**a**) Heat map of unsupervised partitioning of CpG site-specific methylation in localized tumor, metastasized primary tumor, and metastatic tissue samples. Rows show patient/sample-specific relative methylation data with the specified color-coding. The two patient clusters (1,2) shown were obtained by k-means partitioning and consensus clustering with 100-fold bootstrapped resampling. Columns present CpG site-specific methylation data for the indicated genes. The type of analyzed tissue is indicated in the middle heatmap, including localized primary tumors (M0, blue), primary metastatic tumors (M+, orange), and metastatic tissue (Mtx, red), and the right heatmap shows the per sample (row-wise) average methylation (mean M). (**b**) Multi-dimensional scaling analysis of metastatic primary cancer (blue solid circles) and metastatic tissue samples (red solid triangles). (**c**) Principal component analysis of metastatic primary cancer (blue solid circles) and metastatic tissue samples (red solid triangles).

**Figure 6 ijms-23-11190-f006:**
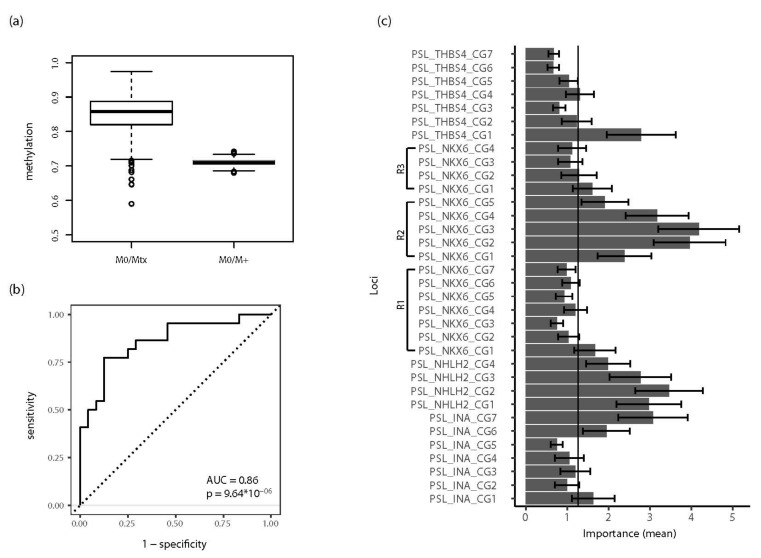
Renal metastasis associated methylation signature (RMAMS) and detection of metastatic primary cancer. (**a**) Box plot analysis of the distributions of ROC-AUC values following 1000 random splits of training and test cohorts and random forest classification analysis for detection of metastatic tissues. (**b**) Exemplary ROC analysis for RMAMS detection of metastatic tissues exhibiting median ROC-AUC of random runs shown in (**a**). (**c**) Bar plot of variables of importance analyses of the random forest classification for the detection of metastatic tissues following 1000 random splits into training and test cohorts. Data are presented as mean and standard deviation (error bars). The vertical line shows the median importance.

**Table 1 ijms-23-11190-t001:** Genomic localization of CpG sites analyzed.

Pyroassay	Gene	Chromosom	TCGA/KIRC Candidate	Genomic Position	AssayCG
	NKX6-2	10	cg06082548	134,598,909	
R3	NKX6-2	10		134,598,942	CG4
				134,598,945	CG3
				134,598,948	CG2
				134,598,952	CG1
R2	NKX6-2	10		134,599,807	CG5
			cg01384488	134,599,809	CG4
				134,599,823	CG3
				134,599,836	CG2
				134,599,841	CG1
R1	NKX6-2	10	cg19701540	134,600,915	CG1
				134,600,919	CG2
				134,600,922	CG3
				134,600,932	CG4
				134,600,934	CG5
				134,600,938	CG6
				134,600,949	CG7

**Table 2 ijms-23-11190-t002:** Hypermethylation of NKX6-2 regions R1-R3 in renal tissues.

Assay	Tumor Specific Hypermethylation	Metastatic Primary Cancer Hypermethylation				Metastatic Tissue Specific Hypermethylation			
	*p*-Value ^1^	Mean Meth. (%)		OR (95% CI)	*p*-Value ^2^	Mean Meth. (%)		OR (95% CI)	*p*-Value ^2^
		M0	M+			M0	Mtx		
R3	6.60 × 10^−17^	14.70	23.08	1.06 (1.03–1.10)	3.37 × 10^−4^	14.14	23.05	1.08 (1.05–1.11)	1.14 × 10^−7^
R2	1.25 × 10^−14^	11.19	17.23	1.04 (1.01–1.07)	0.008	10.46	19.81	1.09 (1.06–1.12)	7.20 × 10^−8^
R1	1.16 × 10^−14^	16.31	23.35	1.04 (1.01–1.07)	0.006	15.82	24.81	1.05 (1.03–1.07)	1.04 × 10^−5^

OR odds ratio; 95% CI 95% confidence interval; M0 primary tumours without metastasis; M+ primary tumours with presence of distant metastases; Mtx metastatic tissue; meth., methylation. ^1^ two-sided paired *t*-test; ^2^ bivariate logistic regression with the covariate age.

## Data Availability

The anonymized datasets used and/or analyzed during the current study are available from the corresponding author upon reasonable request. Due to the General Data Protection Regulation (Art.5 DSGVO), we are not allowed to share sensitive data within an open data-sharing platform.

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
