# Peer review of "Identification of a Novel Renal Metastasis Associated CpG-Based DNA Methylation Signature (RMAMS)"

_ijms, 2022, doi:10.3390/ijms231911190_

Round 1
Reviewer 1 Report
The manuscript “Identification of a Novel Renal Metastasis Associated CpG-based DNA Methylation Signature (RMAMS)” by Jürgen et al. demonstrates interesting and novel results. The Authors assessed whether the detection of metastatic renal tissues using CpG-methylation of INA, NHLH2, and THBS4 (doi:10.3390/cancers14010039) can be improved by including the methylation information of biometrically identified candidate CpG loci within the NKX6-2 gene. The Authors have shown that CpG sites in various regions of the NKX6-2 gene are hypermethylated, in both metastasized primary renal cell cancer (RCC) tissue and RCC-derived distant metastatic tissues. They demonstrated that CpG-based methylation information from INA, NHLH2, NKX6-2, and THBS4 can be used for statistical prediction of the metastatic potential of renal tissues. Their findings support CpG-based methylation as a candidate for molecular stratification of high-risk patients who may benefit from (neo)adjuvant therapy.
Minor comments:
Page 3, Figure 1 legend, line 104: adN abbreviation on the scheme should be added to the legend “tumor-adjacent normal renal tissues (adN)…” – what the star “*” next to “adN*” and “Tumor*” (on the scheme) means?
Page 3, line 117: the gene symbols should be italicized (“the INA, NHLH2, and THBS4 genes”).
Page 7, lines 199, 201, 207, 211 and 214: should it be Table S3?
Table S3: in the Word file with Table S3, there is the caption of this table starting with “Table S2 …”.
The genomic position of CpG sites of the NKX6-2 gene covered by the pyrosequencing assays should be presented.
Page 13, line 424: the Authors’ contributions should be described.
Page 13, line 426: The Supplementary Materials should be listed.
Author Response
The manuscript “Identification of a Novel Renal Metastasis Associated CpG-based DNA Methylation Signature (RMAMS)” by Jürgen et al. demonstrates interesting and novel results. The Authors assessed whether the detection of metastatic renal tissues using CpG-methylation of INA, NHLH2, and THBS4 (doi:10.3390/cancers14010039) can be improved by including the methylation information of biometrically identified candidate CpG loci within the NKX6-2 gene. The Authors have shown that CpG sites in various regions of the NKX6-2 gene are hypermethylated, in both metastasized primary renal cell cancer (RCC) tissue and RCC-derived distant metastatic tissues. They demonstrated that CpG-based methylation information from INA, NHLH2, NKX6-2, and THBS4 can be used for statistical prediction of the metastatic potential of renal tissues. Their findings support CpG-based methylation as a candidate for molecular stratification of high-risk patients who may benefit from (neo)adjuvant therapy.
Minor comments:
Page 3, Figure 1 legend, line 104: adN abbreviation on the scheme should be added to the legend “tumor-adjacent normal renal tissues (adN)…” – what the star “*” next to “adN*” and “Tumor*” (on the scheme) means?
Authors response: We have now added the “adN” abbreviation and removed “*” in adN* and Tumor* labelling in Figure 1
Page 3, line 117: the gene symbols should be italicized (“the INA, NHLH2, and THBS4 genes”).
Authors response: We have corrected the corresponding gene symbols.
Page 7, lines 199, 201, 207, 211 and 214: should it be Table S3?
Authors response: We have checked the relevant section and found no mis-referencing. Data of Figure 4a-c) were statistically evaluated and corresponding results for the three regions R1-R3 were summarized in the three sections of Table 2 (Tumor specific hypermethylation, Metastatic primary cancer hypermethylation, Metastatic tissue specific hypermethylation)
Table S3: in the Word file with Table S3, there is the caption of this table starting with “Table S2 …”.
Authors response: We have corrected description of supplemental table S3
The genomic position of CpG sites of the NKX6-2 gene covered by the pyrosequencing assays should be presented.
Authors response: Table 1 presents genomic positions of all CpG sites covered by the pyrosequencing assays.
Page 13, line 424: the Authors’ contributions should be described.
Authors response: We have revised the author contributions section.
Page 13, line 426: The Supplementary Materials should be listed.
Authors response: We have now added correct information for supplemental materials
Reviewer 2 Report
Well analysis about metastatic risk of renal cell carcinoma in CpG-based methylation information. In the future of precision medicine, how to predict the patient for the best treatment is important. In clinical, the adjuvant therapy about advanced renal cell carcinomausually has some side effects. How to choose the best therapy for patient is very important. In this study, we could realize CpG-based methylation may a candidate for molecular stratification of advanced cancer.
However, in this study, the gene-methylation signaturesite associated with metastatic is INA, NHLH2, NKX6-2, and THBS4. How to explain the different site compare with previous study? Besides, how about the clinical presentation in these gene-methylation signature site?
In conclusion, this article may give the physician more idea about advanced renal cell carcinoma treatment.
Author Response
Comments and Suggestions for Authors
Well analysis about metastatic risk of renal cell carcinoma in CpG-based methylation information. In the future of precision medicine, how to predict the patient for the best treatment is important. In clinical, the adjuvant therapy about advanced renal cell carcinomausually has some side effects. How to choose the best therapy for patient is very important.
In this study, we could realize CpG-based methylation may a candidate for molecular stratification of advanced cancer.
However, in this study, the gene-methylation signaturesite associated with metastatic is INA, NHLH2, NKX6-2, and THBS4.
How to explain the different site compare with previous study?
Authors response: Our previous study made use of unsupervised cluster analysis of CpG sites methylation in the INA,NHLH2 and THBS4 genes (INT) suggesting potential use of INT markers in statistical classification for detection of metastatic cancers. Adding NKX6-2 site methylation information now allowed detection of metastatic tissues and metastatic RCC with good diagnostic efficiency also in supervised analysis using independent test and training cohorts. Moreover, importance plot shows that new information relevant for the classification algorithm is introduced when considering NKX6-2 sites in addition to the INT markers (Figs. 6 and S1). On the other hand, addition of new candidate sites normally change relative importance of former CpG sites in random forest classification. Of note, addition of the NKX6-2 sites did not make the measurement of any of the other previously published gene methylation superfluous
Besides, how about the clinical presentation in these gene-methylation signature site?
Authors response: While our INNT signature clearly allows detection of metastatic RCC tissue, we currently do not have clinical information about late onset of metastasis (metachronous metastasis) in patients for the cohort under analysis. We described in the discussion section that our subsequent studies will aim both on further extension and improvement of the RMAMS methylation signature as well as the setup of a patient cohort allowing evaluation of the predictive power of RMAMS for detection of metachronous metastasis as basis for stratification of high-risk patients for modern adjuvant therapy.
In conclusion, this article may give the physician more idea about advanced renal cell carcinoma treatment.